# ZEBRA: A CONTINUOUS GENERATIVE TRANSFORMER FOR SOLVING PARAMETRIC PDES

**Louis Serrano[1], Pierre Erbacher[1], Jean-Noël Vittaut[2], Patrick Gallinari[1, 3]**
[1] Sorbonne Université, CNRS, ISIR, 75005 Paris, France
[2] Sorbonne Université, CNRS, LIP6, 75005 Paris, France
[3] Criteo AI Lab, Paris, France
`{louis.serrano, pierre.erbacher}@isir.upmc.fr`
`jean-noel.vittaut@lip6.fr, patrick.gallinari@isir.upmc.fr`

## ABSTRACT

Foundation models have revolutionized deep learning, moving beyond task-specific architectures to versatile models pre-trained using self-supervised learning on extensive datasets. These models have set new benchmarks across domains, including natural language processing, computer vision, and biology, due to their adaptability and state-of-the-art performance on downstream tasks. Yet, for solving PDEs or modeling physical dynamics, the potential of foundation models remains untapped due to the limited scale of existing datasets. This study presents Zebra, a novel generative model that adapts language model techniques to the continuous domain of PDE solutions. Pre-trained on specific PDE families, Zebra excels in dynamic forecasting, surpassing existing neural operators and solvers, and establishes a promising path for foundation models extensively pre-trained on varied PDE scenarios to tackle PDE challenges with scarce data.

## 1 INTRODUCTION

Recent advancements in computational science have led to the development of Neural Operators (Lu et al., 2021; Li et al., 2021) and Auto-Regressive Solvers (Brandstetter et al., 2022; Gupta & Brandstetter, 2022; Stachenfeld et al., 2022), which represent significant progress in creating surrogate models for systems governed by partial differential equations (PDEs). These data-driven models, capable of extrapolating system dynamics from novel initial conditions with limited training data, offer an appealing alternative to traditional computational fluid dynamics (CFD) solvers by reducing the dependency on extensive prior physical knowledge. Despite their promise, these models face a significant challenge: they require retraining to adapt to new or slightly modified PDE parameters, indicating a fundamental gap in their generalizability.

One approach to address this challenge involves conditioning models on PDE parameters (Brandstetter et al., 2022; Takamoto et al., 2023; Subramanian et al., 2023). However, this strategy assumes the availability of accurate PDE parameters for all training and test data. An alternative strategy proposes adapting network weights to each specific PDE "environment" (Kirchmeyer et al., 2022), but this approach is hampered by complex meta-training optimizations, limiting scalability and necessitating labels for each environment.

In a tentative to exploit foundation models, McCabe et al. (2023) explored the feasibility of pretraining large models on diverse PDE parameters using a self-supervised approach. They introduced a video-transformer architecture for multiphysics pretraining, demonstrating effective transferability to closely related PDE parameters. Their focus is on training from multiple heterogeneous dynamics instead of solving parametric PDE as we do here.

Our research builds on the foundation laid by self-supervised pretraining successes in both language (Devlin et al., 2018; Radford et al., 2018; 2019; 2021) and vision domains (He et al., 2020; Caron et al., 2021). Given the sequential nature of time-dependent PDE solutions, we investigate whether a generative pretraining approach, similar to that used in language models, could enhance transferability across dynamical systems. Language models, pre-trained on vast and heterogeneous datasets, utilize a probabilistic framework to manage diversity effectively. However, their application to PDEs

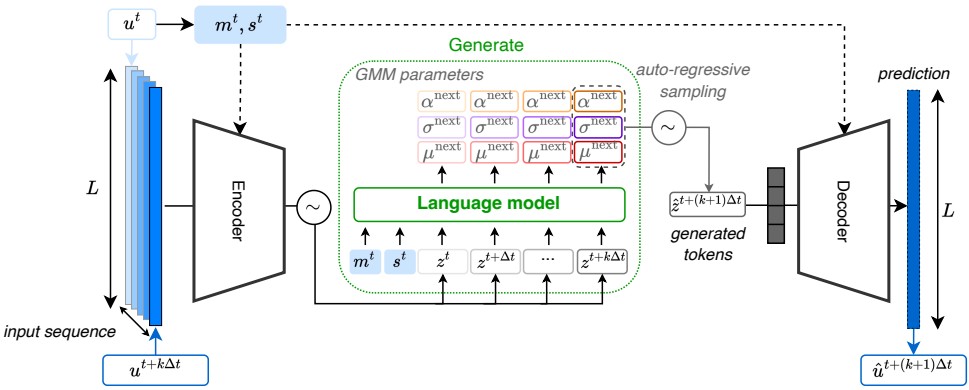

Figure 1: Diagram of Zebra. An encoder compresses the normalized sequence of states, a language model predicts the dynamic and a decoder restores the spatial resolution.

has been limited due to challenges in processing continuous data. Recent efforts to improve numerical tokenization in large language models (Golkar et al., 2023; Gruver et al., 2023) address some of these challenges, yet scalability for physical signals remains a concern. Our contributions are as follows:

- We present Zebra, a novel generative model that adapts language model architectures to the continuous representations of physical states employing a gaussian mixture component, allowing the generation of continuous trajectory distributions.
- We pretrain Zebra and a selection of neural operators/solvers on distinct families of PDEs, evaluating their proficiency in learning diverse dynamics.
- Zebra outperforms all baseline models in auto-regressive rollouts for new trajectories within the distribution of PDE parameters.

## 2 PROBLEM SETTING

We explore the potential of neural solvers and neural operators in capturing the dynamics governed by a spectrum of time-dependent parametric PDEs. Our objective would be to train models that can generalize to multiple parameters including initial conditions, boundary conditions, coefficient parameters, forcing terms. However as a first approach, we focus here on solving PDEs with varying coefficient parameters – such as fluid viscosity or advection speed – broadly denoted by $c$ and keep other aspects of the PDEs similar across dynamics. We define $\mathcal{F}_c$ as the set of PDE solutions corresponding to parameter $c$. A solution $\boldsymbol{u}(x,t)$ within $\mathcal{F}_c$ satisfies:

$$\frac{\partial \boldsymbol{u}}{\partial t} = F\left(c, t, x, \boldsymbol{u}, \frac{\partial \boldsymbol{u}}{\partial x}, \frac{\partial^2 \boldsymbol{u}}{\partial x^2}, \dots\right), \quad \forall x \in \Omega, \forall t \in (0, T] \tag{1}$$

$$\mathcal{B}(\boldsymbol{u})(t, x) = 0 \quad \forall x \in \partial\Omega, \forall t \in (0, T] \tag{2}$$

$$\boldsymbol{u}(0, x) = \boldsymbol{u}^0 \quad \forall x \in \Omega \tag{3}$$

Our goal is to assess the capability of neural architectures to approximate the evolution operator $\mathcal{G}_c : \boldsymbol{u}^t \to \boldsymbol{u}^{t+\Delta t}$ without knowledge of the parameter $c$. Leveraging a sequence of past states $\boldsymbol{u}^{t-k\Delta t:t} := (\boldsymbol{u}^{t-k\Delta t}, \dots, \boldsymbol{u}^t)$ with $k \geq 1$, we aim to train models in a self-supervised manner to forecast $\boldsymbol{u}^{t+\Delta t}$, thereby pretraining across diverse dynamics. We do not give access to the parameters during training to force the models to learn from data the multiple dynamics as in (McCabe et al., 2023). In this work, we will consider only 1D equations with one physical channel.

## 3 MODEL

Our model, called Zebra adopts an encode-generate-decode paradigm, but unlike other models, Zebra makes predictions within a continuous probabilistic framework, as pictured in Figure 1. The architecture is composed of an encoder, a language model and a decoder as described below.

**Encoder** : $\boldsymbol{u}^{0:k} \rightarrow \boldsymbol{z}^{0:k}$. The role of the encoder is to spatially compress the information by reducing the spatial resolution $L$ of the input function into a latent spatial representation with resolution $L^{'} < L$ and channel dimension $d$. We adopt a variational formulation, therefore the outputs of the encoder are multivariate gaussian parameters $\mathcal{E}_w(\boldsymbol{u}_t) = (\boldsymbol{\mu}^t, \boldsymbol{\sigma}^t)$ where $\boldsymbol{\mu}^t = (\mu_1^t, \cdots, \mu_{L^{'}}^t)$, $\boldsymbol{\sigma}^t = (\sigma_1^t, \cdots, \sigma_{L^{'}}^t)$, and from which latent codes can be sampled $\boldsymbol{z}^t \sim \mathcal{N}(\boldsymbol{\mu}^t, (\boldsymbol{\sigma}^t)^2)$. To simplify notations, we will omit $t$ and denote $\boldsymbol{u}^{0:k} = \boldsymbol{u}^{t:t+k\Delta t}$. To be able to efficiently represent a sequence of physical states $\boldsymbol{u}^{0:k}$, which can be of varying amplitudes and include diverse spectrum evolution, into a sequence of codes $\boldsymbol{z}^{0:k}$, we first normalize the input sequence prior to the encoder. We utilize the first observed state in the sequence $\boldsymbol{u}^0$ to compute mean $m^0$ and standard deviation $s^0$ across the spatial dimension and obtain the normalized states as follows: $\tilde{\boldsymbol{u}}^i = \frac{\boldsymbol{u}^i - m^0}{s^0 + \epsilon}$, for $0 \leq i \leq k$. We use a fully convolutional architecture for the encoder.

**Continuous Language Model**: $\boldsymbol{z}^{0:k} \rightarrow \hat{\boldsymbol{z}}^{k+1}$. The role of the language model is to learn and generate the dynamics auto-regressively. It exhibits two key distinctions with standard language model architectures: (i) it takes continuous inputs and (ii) it outputs a continuous probability distribution thus enabling the modeling of continuous trajectory distributions. Our architecture follows the continuous formulation proposed in Tschannen et al. (2024). Given a sequence of latent state representations $\boldsymbol{z}^{0:k} = (\boldsymbol{z}^0, \dots, \boldsymbol{z}^k)$ with $\boldsymbol{z}^i \in \mathbb{R}^{L^{'} \times d}$, the language model returns a continuous probability distribution over the next token $p(\boldsymbol{z}^{k+1}|\boldsymbol{z}^{0:k})$. Our approach utilizes an auto-regressive transformer applying causal attention to process the sequence of states. The sequence $\boldsymbol{z}^{0:k}$, derived from normalized states $\tilde{\boldsymbol{u}}^{0:k}$ is linearly projected to the transformer high-dimensional space. We additionally incorporate a bidirectional attention prefix (Liu et al., 2018) containing the mean $m^0$ and scale $s^0$ tokens projected using two-layer MLPs, $f_m$ and $f_s$ to provide additional context. We use RoPE Su et al. (2021) to encode the relative positions in the sequence.

Therefore, we parameterize the next-token probability distribution $p(\boldsymbol{z}^{k+1}|\boldsymbol{z}^{0:k}, m^0, s^0)$ with $d$-independent gaussian mixture models (GMM). By fixing the number of components in the mixture to $K$, the output of the transformer for each token outside the prefix is a $d \times K$ matrix of mixture probabilities $\boldsymbol{\alpha}^{\text{next}} \in [0, 1]^{d \times K}$, mean parameters $\boldsymbol{\mu}^{\text{next}} \in ]-\infty, +\infty[^{d \times K}$, and scale parameters $\boldsymbol{\sigma}^{\text{next}} \in [0, +\infty[^{d \times K}$. Building on Esser et al. (2021), we transform grid-based prediction into a sequence modeling problem. Therefore, we flatten each latent representations $\boldsymbol{z}^i$ as a sequence of $L^{'}$ tokens of dimension $d$ and sample $\hat{\boldsymbol{z}}^{k+1} \sim p(\boldsymbol{z}^{k+1}|\boldsymbol{z}^{0:k})$, where $p(\boldsymbol{z}^{k+1}|\boldsymbol{z}^{0:k}) = \prod_{l=1}^{L^{'}} p(\boldsymbol{z}_l^{k+1}|\boldsymbol{z}^{0:k}, \boldsymbol{z}_1^{k+1}, \dots, \boldsymbol{z}_l^{k+1})$.

**Decoder**: $\hat{\boldsymbol{z}}^{k+1} \rightarrow \hat{\boldsymbol{u}}^{k+1}$. The decoder maps the latent tokens $\hat{\boldsymbol{z}}^{k+1}$ to the physical space $\hat{\boldsymbol{u}}^{k+1} = \mathcal{D}_\psi(\hat{\boldsymbol{z}}^{k+1})$. We denormalize with $m^0$ and $s^0$ to get the reconstruction.

**Inference** Given an encoded sequence $\boldsymbol{z}^{0:k}$, predicting the latent representation $\boldsymbol{z}^{k+1}$ requires predicting $L^{'}$ tokens autoregressively. Each token $\boldsymbol{z}_l^{k+1}$ with $1 < l < L^{'}$ is sampled from the K-GMM distribution $\boldsymbol{z}_l^{k+1} \sim \text{GMM}(\boldsymbol{\alpha}_l^{k+1}, \boldsymbol{\mu}_l^{k+1}, \boldsymbol{\sigma}_l^{k+1})$ and added to the sequence. To unroll the dynamics, we use a rolling window of constant size, meaning that the prediction $\hat{\boldsymbol{u}}^{k+1}$ is made using an initial sequence $\boldsymbol{u}^{0:k}$, the following prediction $\hat{\boldsymbol{u}}^{k+2}$ is made using $(\boldsymbol{u}^{1:k}, \hat{\boldsymbol{u}}^{k+1})$.

**Training** We do a two-stage training. We first train the encoder-decoder as a variational auto-encoder (VAE) Kingma & Welling (2014) and then train the language model on the latent representations obtained with the encoder. We provide further details on training in the sections below.

### 3.1 FIRST STEP: VAE TRAINING

The encoder and decoder are jointly trained as a VAE. We use a mixed loss objective:

$$\mathcal{L} = \mathcal{L}_{\text{recon}} + \beta \cdot \mathcal{L}_{KL} \tag{4}$$

with $\beta > 0$; where $\mathcal{L}_{\text{recon}} = \sum_{i=0}^{i=k+1} \frac{\|\boldsymbol{u}^i - \hat{\boldsymbol{u}}^i\|_2}{\|\boldsymbol{u}^i\|_2}$ is the relative L2 loss between the input $\boldsymbol{u}^i$ and its reconstruction $\hat{\boldsymbol{u}}^i$ through the encoder-decoder. The KL term $\mathcal{L}_{\text{KL}} = \sum_{i=t}^{i=k+1} D_{\text{KL}}(\mathcal{N}(\boldsymbol{\mu}^i, (\boldsymbol{\sigma}^i)^2) \,||\, \mathcal{N}(0, I))$ regularizes the network.

## 3.2 SECOND STEP: LANGUAGE MODEL TRAINING

Zebra is trained using self-supervised learning on next token prediction task (Radford et al., 2018) using teacher forcing. Given the sequence $z^{0:k+1}$ and normalization scalars $m^0, s^0$, the model is trained to minimize the negative log-likelihood (see Appendix A) expressed below:

$$\mathcal{L}_{\text{nll}} = -\sum_{i=1}^{k+1} \log p(z^i | z^0, \dots, z^{i-1}, m^0, s^0) \tag{5}$$

## 4 EXPERIMENTS

**Datasets**  We assess the model's performance on 1D data, leaving more complex tests to further investigations. We used the 1D Advection (*Advection*) and 1D Burgers (*Burgers*) equation datasets from PDEBench (Takamoto et al., 2022), training across all available PDE parameters. For *Advection*, the coefficient parameter that varies across equations is the advection speed $\beta \in \{0.1, 0.2, 0.4, 0.7, 1.0, 2.0, 4.0, 7.0\}$ and for *Burgers* it is the fluid viscosity $\nu \in \{0.001, 0.002, 0.004, 0.01, 0.02, 0.04, 0.1, 0.2, 0.4, 1.0, 2.0, 4.0\}$. We evaluate all models on in-distribution parameters. Namely, for each parameter, we have access to 10,000 trajectories, generated with different initial conditions, that we split in 95% for training, and 5% for testing. The trajectories were generated with a spatial resolution of 1024 and with a temporal resolution of 200, which we downsample to 256 and 100 respectively.

**Baselines**  We compare our model to current state-of-the-art neural architectures: ● Fourier neural operator (FNO) (Li et al., 2021) ● UNet (Gupta & Brandstetter, 2022) ● ResNet (Gupta & Brandstetter, 2022).

**Training and evaluation**  To make a fair comparison, we pretrain all baselines with teacher-forcing, i.e. with 1-step prediction by learning to map $(u^t, \dots, u^{t+k\Delta t}) \rightarrow u^{t+(k+1)\Delta t}$. We fix $k = 4$ for all models and all datasets. We use the relative L2 loss for training. We evaluate the pretraining in a zero-shot setting on in-distribution new trajectories. We rollout the models for 1, 10, and 25 steps starting from solutions at different times ($t = 0, t = 25, t = 50, t = 70$) and compute the relative L2 loss with the ground truth trajectories.

**Results**  The results in Figure 2 show that all the models interpolate up to some extent to the PDE parameters distribution. Our model outperforms the baselines on *Advection* and *Burgers*. When evaluating the performance at 1, 10, and 25 rollouts, Zebra consistently demonstrates lower error rates, indicating its enhanced ability to accurately predict and model multiple dynamics. Our model has been designed to be easily adaptable to multiple settings such as variable spatial and temporal resolutions, diverse context sizes, etc.

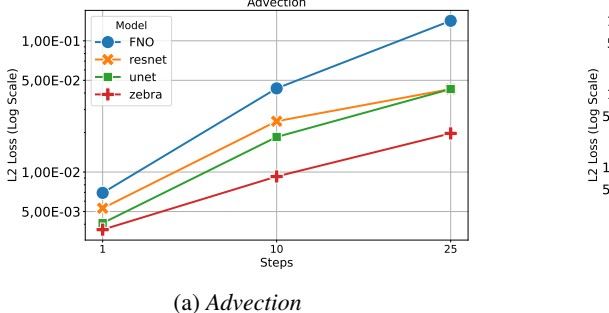
(a) *Advection*

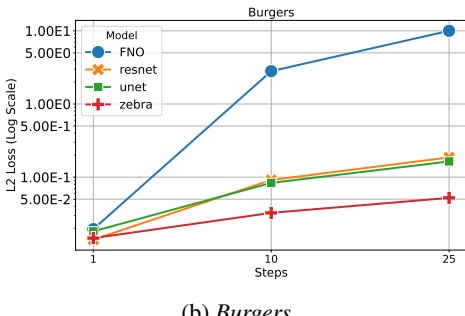
(b) *Burgers*

Figure 2: Test relative L2 loss for rollouts of lengths $(1, 10, 25)$, starting from $t = 0$.

## 5 CONCLUSION

This study presents Zebra, a novel generative model that adapts language model techniques to the continuous domain of PDE solutions, demonstrating enhanced learning and generalization across diverse dynamics. By effectively pretraining on a family of PDE, Zebra's performance in autoregressive rollouts surpasses all baseline models, underscoring its effectiveness in generating accurate new trajectories within the PDE parameter distribution, thus offering a promising approach to overcome scalability in PDE applications.

## ACKNOWLEDGEMENTS

We acknowledge the financial support provided by DL4CLIM (ANR-19-CHIA-0018-01), DEEP-NUM (ANR-21-CE23-0017-02), PHLUSIM (ANR-23-CE23-0025-02), and PEPR Sharp (ANR-23-PEIA-0008, ANR, FRANCE 2030). This work was performed using HPC resources from GENCI–IDRIS (Grant AD011013522R1).

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

## A   ARCHITECTURE DETAILS

**Likelihood of K-GMM**   In the following we write the likelihood of Gaussian mixture model with $K$ components.

$$p(X|\theta) = \prod_{n=1}^{N} \left( \sum_{k=1}^{K} \alpha_k \frac{1}{\sqrt{2\alpha\sigma_k^2}} \exp\left(-\frac{(x_n - \mu_k)^2}{2\sigma_k^2}\right) \right) \tag{6}$$

$$L = \log p(X|\theta) = \sum_{n=1}^{N} \log \left( \sum_{k=1}^{K} \alpha_k \frac{1}{\sqrt{2\alpha\sigma_k^2}} \exp\left(-\frac{(x_n - \mu_k)^2}{2\sigma_k^2}\right) \right) \tag{7}$$

with $X$ being modeled as a mixture of $K$ Gaussian distributions. Each Gaussian distribution $k$ has a mean $\mu_k$, variance $\sigma_k^2$, and mixture coefficient $\alpha_k$

**Variational auto-encoder**   We provide a schematic view of the VAE framework in Figure 3. We use a spatial encoder and decoder with convolutions only, and normalize the input sequence before the encoder and denormalize to get the final reconstructions. We use normalization statistics computed spatially with the first observation of the sequence.

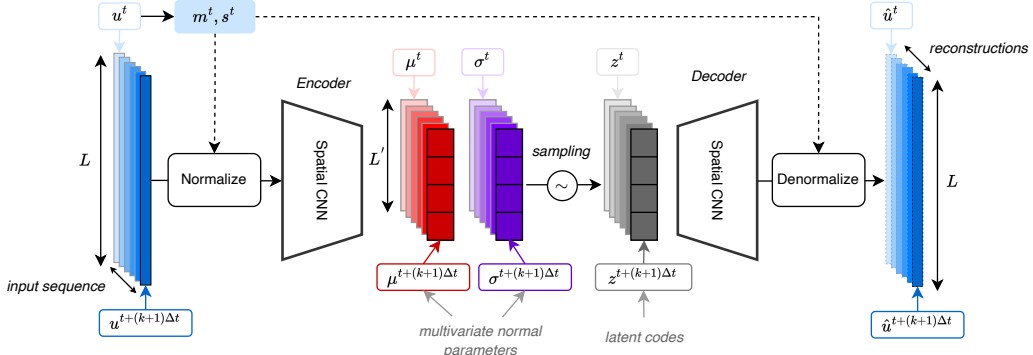

Figure 3: Architecture of the VAE

**Transformer**   We illustrate the components of our continuous transformer in Figure 4. We linearly project the codes obtained with the encoder, and add a prefix with the tokens derived from the normalization statistics. We discard the prefix from the outputs of the causal transformer and then apply a mean head, a scale head, and a mixture head to obtain the GMM coefficients.

## B   RELATED WORK

**Language model for images and videos**   Several works have considered language modeling for image or video generation by combining a VQ-VAE (Oord et al., 2017) with a generative language transformer (Brown et al., 2020). For images, VQGAN Esser et al. (2021) improved upon this framework by using a perceptual and an adversarial loss to boost the reconstructions quality of the decoder from the quantized latent representation. Instead of employing a causal transformer,

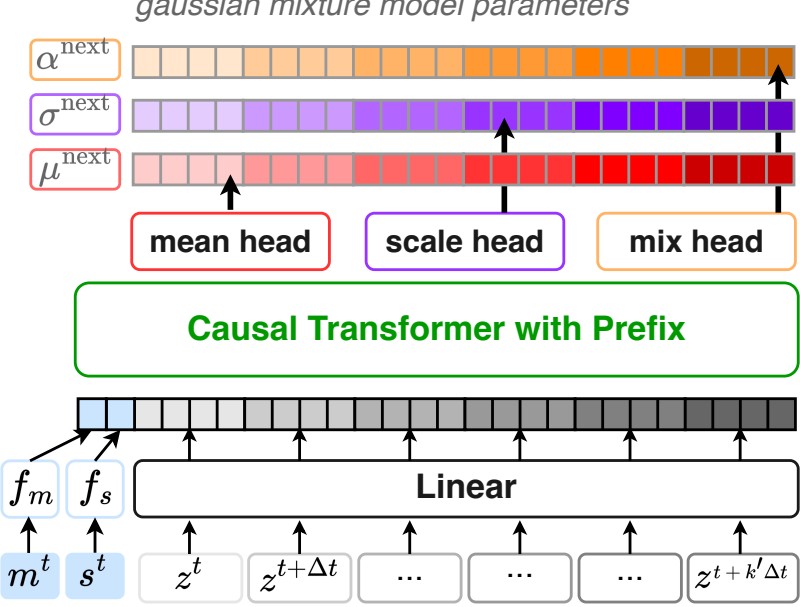

Figure 4: Architecture of our continuous causal transformer.

MaskGIT proposed a bi-directional transformer (Devlin et al. (2018)) with a scheduling mechanism to accelerate the sampling procedure. Departing from this direction, Tschannen et al. (2024) recently introduced a continuous transformer for image generation and assessed its performance with the causal and masked architectures. In video generation, similar techniques relying on quantization have also been introduced, first with VideoGPT (Yan et al., 2021), and more recently with magvit (Yu et al., 2023a) and magvit2 (Yu et al., 2023b). The latter, which introduced a new quantization scheme, obtained results on-par with diffusion models. To the best of our knowledge, there is no continuous transformer for video generation.

**Multiple Physics**   Recent works have advanced neural solvers and surrogate models for PDEs within Scientific Machine Learning (SciML). Brandstetter et al. (2022) introduced a message-passing neural solver that includes additional parameter inputs to improve its time-stepping scheme. Takamoto et al. (2023) developed a channel attention mechanism, using parameter embeddings of PDE coefficients to boost the generalization of neural surrogate models across different PDE parameters. Subramanian et al. (2023) investigated the application of pre-trained ML models to SciML via transfer learning, finding that fine-tuning these models can significantly reduce the necessity for extensive downstream examples for adaptation.

Kirchmeyer et al. (2022) addressed the issue of unknown PDE parameters at the time of inference by adjusting network weights to better suit observed dynamics, using meta-learning to adapt the network across varied environments. Extending the concept of adaptability, McCabe et al. (2023) delved into multiphysics modeling with foundation models. They proposed an architecture to integrate diverse physical signals from different PDEs into a single high-dimensional space, applying a transformer architecture, similar to those in video prediction, to forecast dynamics. Their method, especially effective for 2D equations, highlights the potential of using foundation models in for modeling PDEs and more broadly for SciML applications.

## C    ADDITIONAL RESULTS

### C.1    BURGERS FULL RESULTS

The Burgers' equation under consideration lacks a forcing term, leading to a rapid decrease in fluid velocity amplitude over time. This aspect makes the dynamics challenging to predict at the early stages of the trajectory for all models, a phenomenon highlighted in Figure 5. For instance, FNO is not able to unroll the dynamics from $t = 0$, while it does not diverge when starting from a later timestamp. In contrast, our model is capable of handling this dynamics quite well, as it consistently outperforms other baselines, for all the considered starting times (Figure 6).

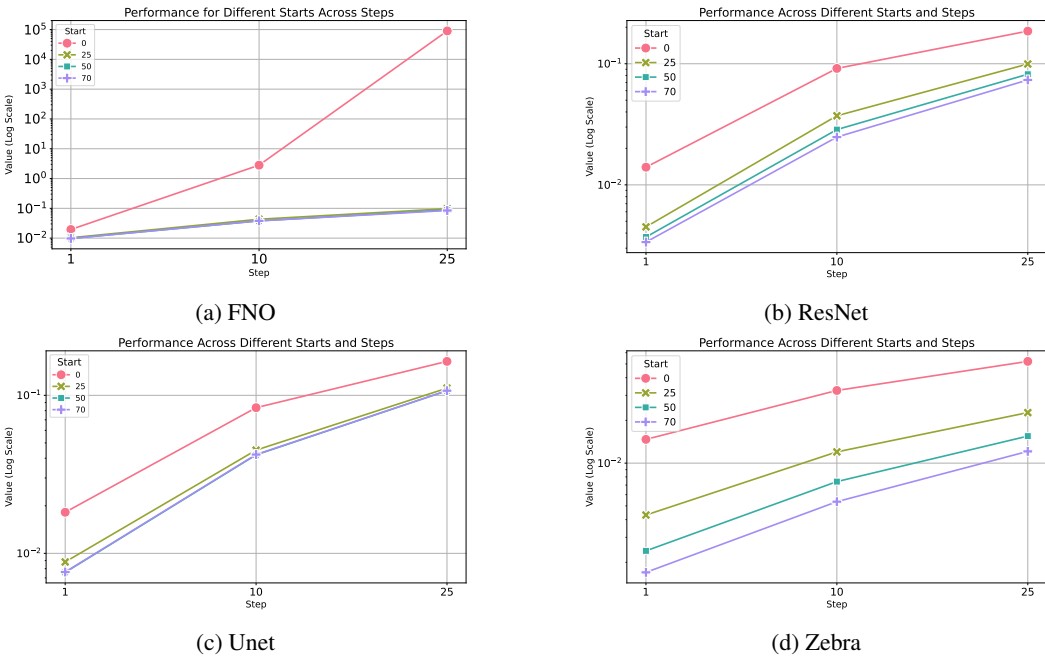

Figure 5: *Burgers* - Starting time influence on each model - Test relative L2 loss with different models for rollouts of lengths $(1, 10, 25)$, starting from $(t = 0, t = 25, t = 50, t = 70)$.

### C.2    ROBUSTNESS TO THE CONTEXT SIZE

In the experiments, we fixed $k = 4$, which means the baselines have been trained and tested with an input sequence of size 5. They cannot be used outside of this scope, with less or more context. In contrast, our architecture benefits from the flexibility of transformers and can work with different context lengths. In theory, it should be enough to infer the correct PDE coefficient parameter from the observation: $(\boldsymbol{u}^t, \boldsymbol{u}^{t+\Delta t})$ and thus to unroll the correct dynamics. However, in practice this depends on the nature of the equation, and providing more context is useful to decrease the uncertainty over the estimation. Experimentally, we observed that our model performed at the same level for *Advection* with a context of size 2, better than all the baselines, and that its performance slightly deteriorated for *Burgers*, though still outperforming other baselines. We show the results in Figure 7

### C.3    QUALITATIVE RESULTS

We provide a comparison of the predicted trajectories with Zebra and the ground truth for several test trajectories on *Advection* with $\beta$=2.0 (Figure 8, Figure 9, Figure 11, Figure 10, Figure 12) and *Burgers* with $\nu = 0.01$ (Figure 13, Figure 14, Figure 15, Figure 16, Figure 17). In both cases, we start from the initial condition at $t = 0$ and unroll for 25 steps. For clarity, we represent only one timestamp out of 5 for advection.

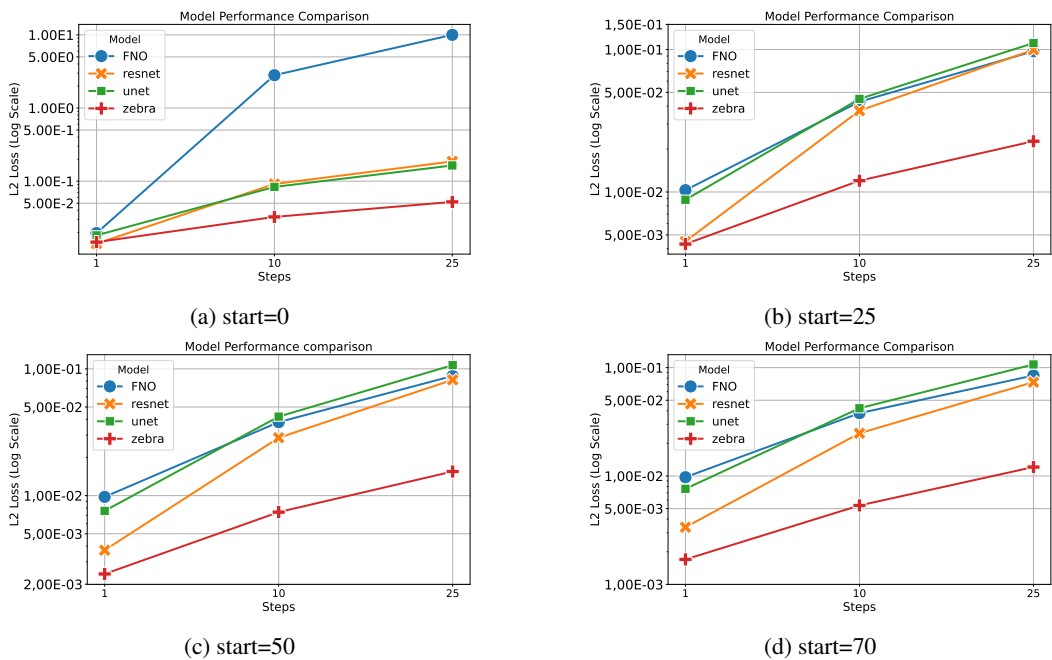

Figure 6: *Burgers* - Model Comparison by starting time - Test relative L2 loss for rollouts of lengths $(1, 10, 25)$, starting from $(t = 0, t = 25, t = 50, t = 70)$.

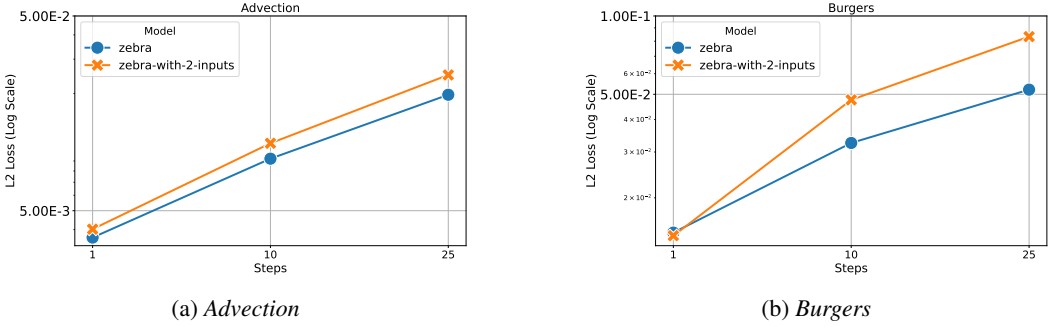

Figure 7: Comparative performance between Zebra with an input sequence of size 5, as used during the pretraining, and a sequence of size 2. We start from $t = 0$ and unroll for 1, 10, 25 steps.

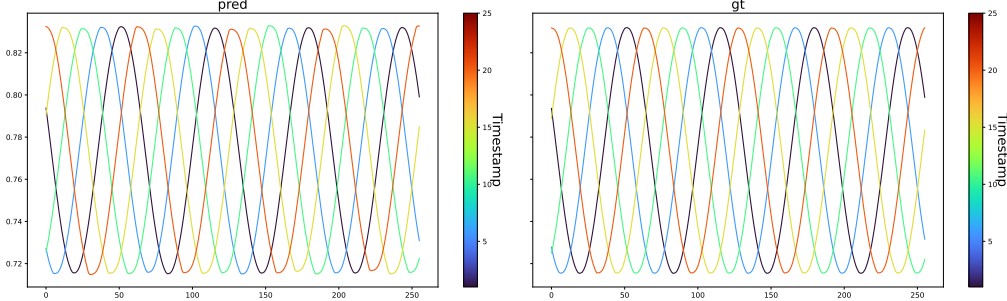

Figure 8: Test trajectory on *Advection* ($\beta = 2.0$). Left is the predicted trajectory and right is the ground truth.

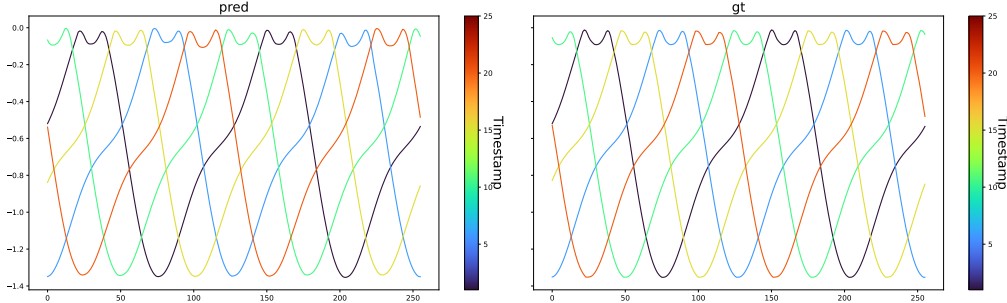

Figure 9: Test trajectory on *Advection* ($\beta = 2.0$). Left is the predicted trajectory and right is the ground truth.

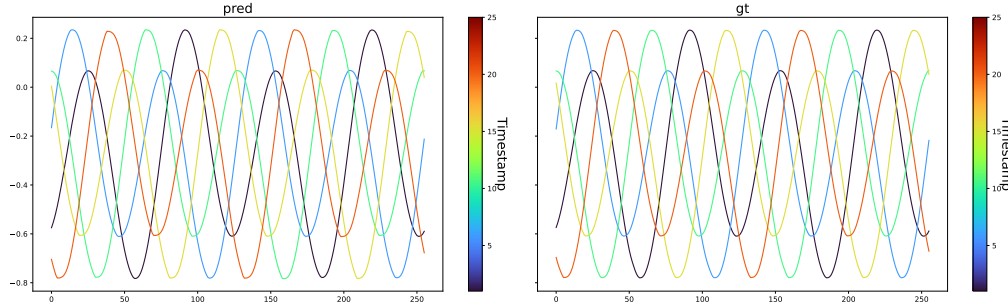

Figure 10: Test trajectory on *Advection* ($\beta = 2.0$). Left is the predicted trajectory and right is the ground truth.

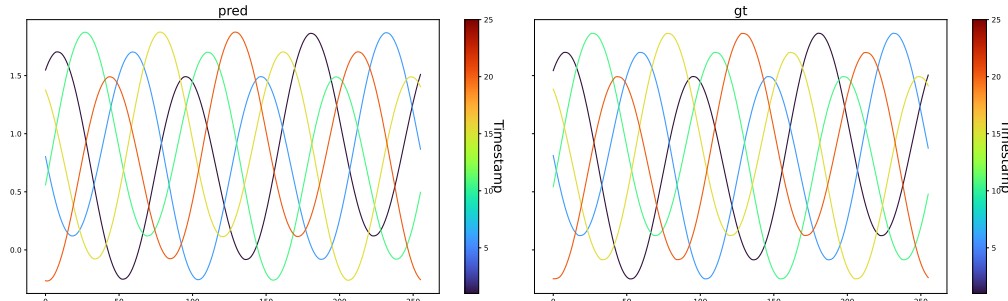

Figure 11: Test trajectory on *Advection* ($\beta = 2.0$). Left is the predicted trajectory and right is the ground truth.

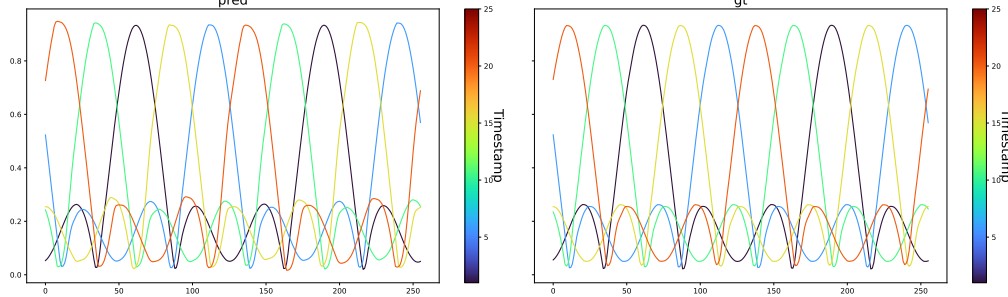

Figure 12: Test trajectory on *Advection* ($\beta = 2.0$). Left is the predicted trajectory and right is the ground truth.

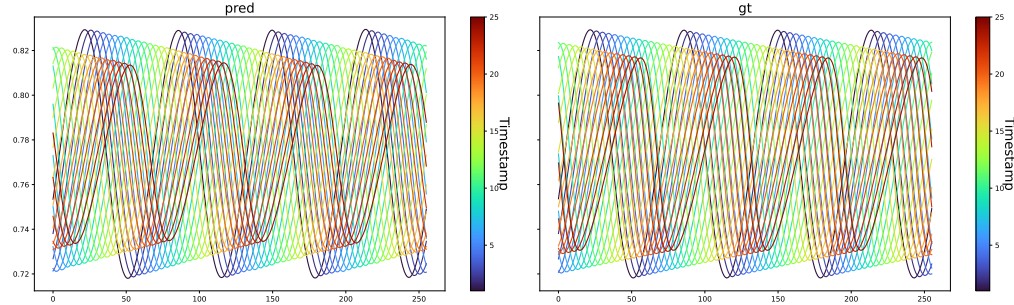

Figure 13: Test trajectory on *Burgers* ($\nu = 0.01$). Left is the predicted trajectory and right is the ground truth.

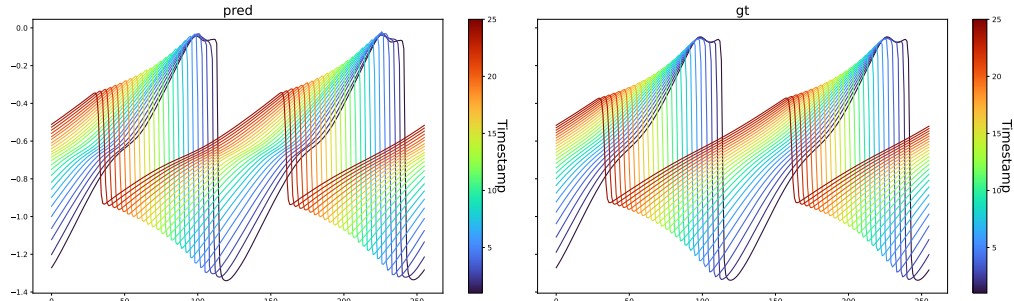

Figure 14: Test trajectory on *Burgers* ($\nu = 0.01$). Left is the predicted trajectory and right is the ground truth.

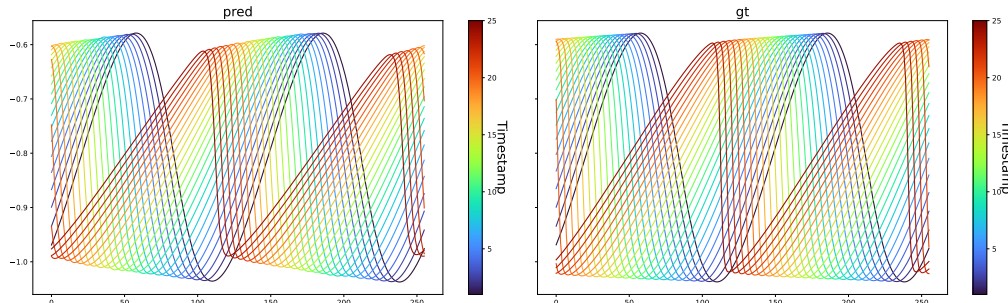

Figure 15: Test trajectory on *Burgers* ($\nu = 0.01$). Left is the predicted trajectory and right is the ground truth.

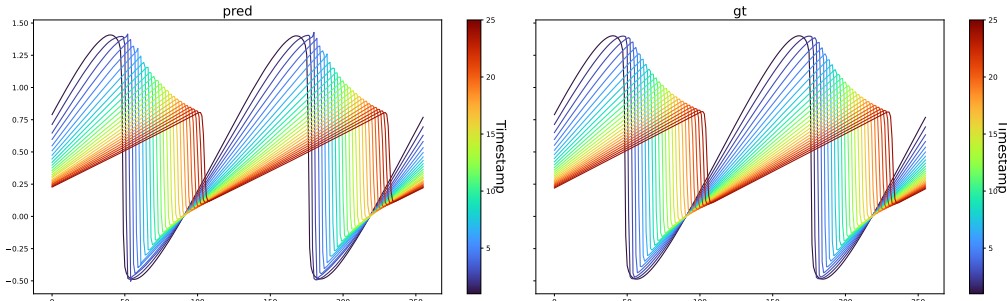

Figure 16: Test trajectory on *Burgers* ($\nu = 0.01$). Left is the predicted trajectory and right is the ground truth.

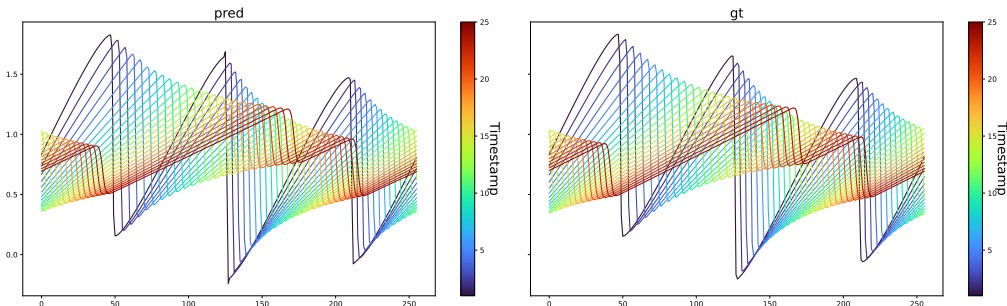

Figure 17: Test trajectory on *Burgers* ($\nu = 0.01$). Left is the predicted trajectory and right is the ground truth.

