# OpenReview forum: "Zebra: a continuous generative transformer for solving parametric PDEs"
_ICLR.cc/2024/Workshop/AI4DiffEqtnsInSci — AI4DiffEqtnsInSci @ ICLR 2024 Poster_

### Official Review · Reviewer_SsUG · 2024-02-24
**Zebra: a continuous generative transformer for solving parametric PDEs**

**Rating:** 7
**Confidence:** 5

**Review:**

Zebra introduces a probabilistic framework for modeling PDEs taking inspiration from the language model community in NLP.


**Strengths**:
- A new approach, from the prism of generative model, to train foundation model for PDEs.
- No explicit incorporation of PDE parameter values in the model.
- Yet another way to perform the *latent evolution* of PDE dynamics.
- Natural support for continuous (a.k.a. discretization-free) modeling.


**Weaknesses**:
- No separate *validation* set. Hence, the *test* set is somewhat "seen" by the model.
- Generalization ability to unseen parameter distribution (OOD) is not tested.
- No experiments on 2D and 3D PDEs, making the generalizability of the proposed approach to higher dimensions uncertain.

**Questions**:
- For the benefit of the broader SciML community, will the codebase be made public upon acceptance?

> We pretrain Zebra and a selection of neural operators/solvers on distinct families of PDEs, evaluating their proficiency in learning diverse dynamics.

- Were both the Advection and Burgers' datasets combined and a single Zebra model was trained? If yes, how was the specific PDE identified?

**Suggestions for improvement**:
- Make notations in Figure 1 consistent with the Problem Setting in Section 2 or vice-versa.

---

### Official Review · Reviewer_NKww · 2024-02-27
**A clean and well-motivated paper that would benefit from more extensive experiments and more challenging comparisons.**

**Rating:** 6
**Confidence:** 3

**Review:**

Summary:
The authors propose to utilize a continuous formulation of generative pre-trained Transformers for PDE forecasting. The continuous formulation is achieved by predicting the parameterizing a Gaussian Mixture Model. Additionally, the computational complexity is reduced by operating in the latent space of a pretrained VAE.
Experimental evaluation is performed on the 1d Burgers and Advection families of PDEs and compares favorably to previously published baseline methods.

Pros:
- Experimental results support consistent improvements over the baselines.
- The framework appears to be of general interest and utility beyond what is shown in this work.
- The text is easy to follow and clearly written.
- The approach is well-motivated from the related work and context of the field.

Cons:
- Technical novelty appears to be low, the authors cite several studies that essentially showed the utility of all components used here.
- Experimental evaluation is performed only on rather simple case studies, precluding any conclusions about the applicability of the framework to more relevant settings.
- The model is only compared against (commonly used) baselines, but not against numerical solvers or more advanced DL methods.
- Performance is only evaluated in terms of L2 loss.

Minor:
- The term "foundation model" is used inflationary here, no scaling properties of the model are shown, nor zero-shot performance on other families of PDE.
- A few language hickups (e.g. 'In a tentative to exploit foundation models')
- Out of curiosity: where does the model name come from?

---

### Meta-Review · Area_Chair_nGXb · 2024-02-29

**Recommendation:** Accept (Poster)

**Metareview:**

This study presents Zebra, a novel generative model that adapts language model techniques to the continuous domain of PDE solutions, which can mitigate the limitation of dataset. I believe this paper can bring new insight to the community.  It is expected that authors will be addressing comments by the reviewers in the final draft.

---

### Decision · Program_Chairs · 2024-02-29

Accept (Poster)